# UV-A for Tailoring the Nutritional Value and Sensory Properties of Leafy Vegetables

Kristina Laužikė *[ID], Akvilė Viršilė [ID], Giedrė Samuolienė [ID], Rūta Sutulienė and Aušra Brazaitytė [ID]

Lithuanian Research Centre for Agriculture and Forestry, Institute of Horticulture, Kauno Str. 30, Kaunas dist., LT-54333 Babtai, Lithuania; akvile.virsile@lammc.lt (A.V.); giedre.samuoliene@lammc.lt (G.S.); ruta.sutuliene@lammc.lt (R.S.); ausra.brazaityte@lammc.lt (A.B.)
* Correspondence: kristina.lauzike@lammc.lt

**Abstract:** This study aims to expand the artificial lighting potential of controlled environment cultivations systems by introducing UV-A (~315–400 nm) wavelengths into the traditional, visible spectrum lighting, seeking to improve the nutritional and sensory value of cultivated leafy vegetables. The experiment was conducted in a closed climate-controlled chamber, maintaining 21/17 °C day/night temperature, ~55% relative humidity, and a 16 h photo/thermo period. Several genotypes of leafy vegetables, red and green leaf lettuce cultivars, mustard, and kale were cultivated under 250 µmol m$^{-2}$s$^{-1}$ basal LED lighting, supplemented by 385 nm UV-A or 405 UV-A/violet wavelengths for 1.1 mW cm$^{-2}$ for 12 h photoperiod for the whole cultivation cycle. The results show that UV-A/violet light impacts on leafy vegetable growth, free radical scavenging activity, sugar, and phytochemical (α tocopherol, α + β carotenes, epicatechin, rosmarinic and chicoric acid contents) are species-specific, and do not correlate with untrained consumer's sensory evaluation scores. The 405 nm light is preferable for higher antioxidant and/or sensory properties of kale, mustard, and green leaf lettuces, but both UV-A wavelengths reduce growth parameters in red leaf lettuce.

**Keywords:** light; lettuce; mustard; kale; sensory properties; phytochemicals

## 1. Introduction

The complex of shrinking agricultural areas, changing climate, and increasing consumer demand is shifting the cultivation of leafy vegetables into precision-controlled environment agriculture (CEA). Not only consumers but also growers are paying more attention to the quality of leafy vegetables. The first impression of the consumer and his choice is usually based on the shape and appearance of the vegetable. However, there is a growing tendency in consumer preferences that higher organoleptic properties of vegetables should go hand in hand with higher nutritional and functional properties [1–3]. External quality, leaf coloring, morphology, and other organoleptic properties do not always coincide with high internal quality indicators [4].

Adequate artificial lighting is necessary to achieve all these goals when growing plants in CEA. LEDs (Light emitting diodes) are getting widely applied for plant lighting; they are characterized by high photon efficiency, and their prices are becoming more accessible, which is of primary relevance for growers [5,6]. Research aiming to improve the growth and quality of leafy vegetables using LED technologies has been performed for over two decades. It was shown that the red and blue LED spectra combination is sufficient for diverse plant growth and photosynthesis; as the highest photon efficacy characterizes these LED colors, it allows to reduce cultivation and investment costs [7–9]. Studies show that blue and red diodes combined in various ratios can increase plant growth, accumulative dry matter content, levels of antioxidant compounds as well as other beneficial substances in the leaves of leafy vegetables [10–12]. Along with blue and red or white LED colors in the basal lighting spectra, supplemental UV-A (315–400 nm) LED light is recently also engaged

in research [10,13,14]. About 6% of solar radiation at sea level in nature includes UV; UV-A radiation accounts for ~95% of solar UV radiation [15,16]. Applying UV-A LEDs in artificial lighting spectra in CEA is not yet common; however, research data show, that supplemental UV-A light can positively affect different plants. It was reported that supplemental UV-A light resulted in bigger biomass, leaf area, pigment, antioxidant, and aliphatic glucosinolate contents in Chinese kale [17–19]; higher shoot biomass, antioxidant phenolics and sugar alcohols in ice plants [20]; higher biomass [21], anthocyanin and ascorbic acid [21], as well as increased protein concentration, mineral nutrients [22] in lettuce. Though showing comparable general trends, studies report species, UV-A dosage, and UV-A wavelength-specific impacts [23–26]. Differential experimental conditions and basic lighting properties aggravate the comparison of the results. Moreover, there is a lack of studies on the impact of UV-A light on the organoleptic properties of leafy vegetables and consumer preferences.

Few UV studies involve food plants, and their results are controversial. Verdaguer et al. [23] analysis of UV studies show differences in UV effects on biomass, morphology, and photosynthesis, which differ depending on the plants. However, there is still a lack of information on the effects of UV-A on the nutritional and commercial quality of leafy vegetables. The prospects for applying UV-A lighting in precision CEA cultivation systems are closely dependent on the development of UV-A LED technology and the accumulated detailed knowledge of the physiological response of plants to the applied UV-A light parameters. Therefore, this study aimed to evaluate and compare the effect of supplemental UV-A LED light on the nutritional quality and organoleptic properties of leafy vegetables: mustard, kale, and several lettuce varieties.

## 2. Materials and Methods

### 2.1. Plant Material and Growing Conditions

Experiments were performed in controlled environment cultivation chambers. Day/night temperatures of $21/17 \pm 2$ °C, 16 h photoperiod, relative humidity of ~55%, and the essential photosynthetic photon flux density (PPFD) of 250 $m^{-2}s^{-1}$ were maintained. PPFD was measured at the plant top level using a photometer–radiometer (RF-100, Sonopan, Poland). Basic lighting was provided by four channel controllable light emitting diode (LED) lighting units (TUAS GTR 2V 0021096109 C1 DL ST, Tungsram, Budapest Hungary), maintaining the spectral composition of deep red 61%, blue 20%, white 15%, and far red 4%.

Three lighting treatments were performed (Figure 1): the basic LED lighting spectra ere supplemented with 385 nm LED UV-A light. Its impact was compared to 405 nm supplemental LED light, which is on the boundary of visible violet and UV-A light. The LED light wavelengths available commercially were selected according to their sufficient efficiency and ageing parameters. Custom-made lighting units, consisting of LEDs, emitting at 385 (LZI-10UB0R-00U4, Osram Sylvania Inc., Wilmington, United States) and 405 nm (LZ1-10UB0R-00U8, Osram Sylvania, Wilmington, United States) were used for supplemental lighting; irradiance of 1.1 mW $cm^{-2}$ (11 W $m^{-2}$) for 12 h photoperiod was measured and maintained at plant top level (FLAME-S-UV-VIS-ES spectrometer (Ocean Optics, Ostfildern, Germany)).

Three varieties of lettuce (*Lactuca sativa*) were grown in this experiment: Affl­cion (green, crinkled, curly leaves), Lollo Rossa (green to pink crinkled, curly leaves), Belino (green to dark red flat leaves) (Sėklos.lt, Lithuania), as well as kale (*Brassica oleracea*) Cavolo Nero di Toscana (dark green leaves) and mustard (*Brassica juncea*) Red Lion (green to reddish leaves) (CN seeds, Ely, UK). Seeds were germinated in water-soaked rockwool cubes (20 × 20 mm; Grodan, Roermond, The Netherlands) and placed under basic lighting spectrum, PPFD of 250 μmol $m^{-2}s^{-1}$. Ten days after germination, seedlings were transferred into deep water culture (DWC) 40 L volume hydroponic tanks; each DWC tank contained 12 mesh pots, and supplemental UV-A lighting treatment was started. The experiment was performed in three replications per each lighting treatment (one replicate–one DWC tank). Nutrient solution concentrate was provided by Plagron (Hydro A&B, Ommelpad, The Netherlands). The pH values were maintained at 6.5 throughout all cultivation pe-

riod. After the plants reached technical maturity, measurements and sample collection were performed. All measurements in each experimental replication were performed in 3 analytical replications (*n* = 9). Plant material was lyophilized (FD-7, SIA Cryogenic and Vacuum Systems, Ventspils, Latvia), ground and dry plant material was used for biochemical analyses.

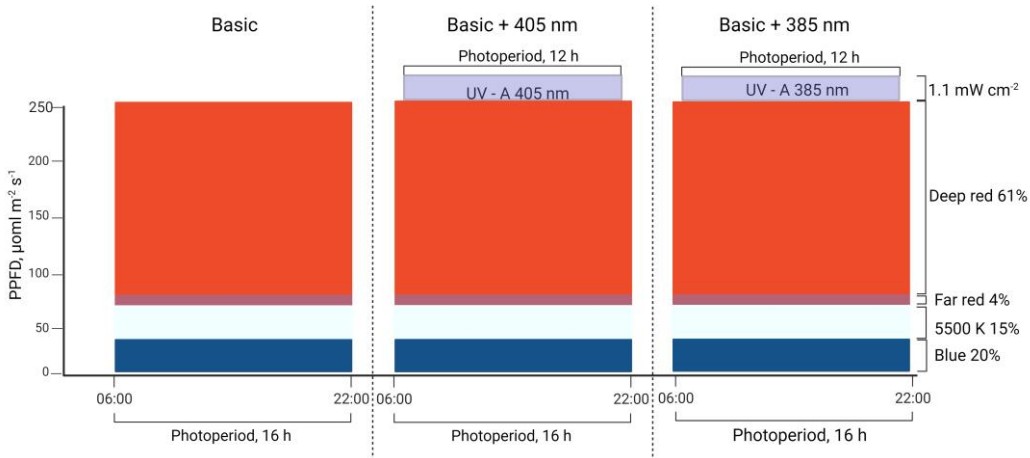

**Figure 1.** Experimental lighting design.

### 2.2. Sensory Analysis

The sensory parameters of the fresh leafy greens' appearance, taste, crispness, bitterness, and sweetness were surveyed to evaluate consumer preferences. A scale of 0–5 was used: 5—maximal score (the sweetest, the most bitter, etc.), 3—neutral, and 0—the least score. Random samples from all three treatment treatments were evaluated by 12 untrained consumer panellists aged 25–65 years (equal gender ratio).

### 2.3. Determination of Soluble Sugars by UFLC (Ultra Fast Liquid Chromatography) Method

Soluble sugar contents were evaluated using the UPLC method with evaporative scattering detection (ELSD). About 0.05 g of lyophilized plant tissue was homogenized and diluted with 4 mL of deionized water. The extraction was carried out for 4 h at room temperature and centrifuged at $14,000 \times g$ for 15 min. A cleanup step was performed before the chromatographic analysis: 1 mL of the supernatant was mixed with 1 mL 0.01% (*w:v*) ammonium acetate in acetonitrile and incubated for 30 min at 4 °C. After incubation, samples were centrifuged at $14,000 \times g$ for 15 min and filtered through a 0.22 µm syringe filter, nylon (BGB Analytik, Boeckten, Switzerland). Analysis was performed on Shimadzu Nexera (Japan) system. Separation was performed on a Supelcosil $250 \times 4$ mm $NH_2$ column (Supelco, Pennsylvania, USA) using 77% acetonitrile as the mobile phase at 1 mL min$^{-1}$ flow rate. A calibration method was used for sugar quantification (mg g$^{-1}$ in dry plant weight (DW)).

### 2.4. Determination of Carotenes by HPLC (High-Performance Liquid Chromatography) Method

The concentration of α and β carotenes were evaluated by HPLC on the YMC Carotenoid column (3 µm particle size, $150 \times 4.0$ mm) (YMC, Kyoto, Japan). Extraction was performed using 80% aqueous acetone (0.05 g lyophilized material homogenized and diluted with 5 mL acetone solution; extraction was carried out for 24 h at 4 °C), then centrifuged (5 min, $4000 \times g$) and filtrated through a 0.22 µm nylon membrane syringe filter (BGB Analytik, Boeckten, Switzerland). An HPLC 10A system (Shimadzu, Kyoto, Japan) with a diode array (SPD-M 10A VP) detector was used for analysis. Peaks were detected at 440 nm. The identification and quantification of carotenes were carried out according to standard materials. The mobile phase consisted of A (80% methanol, 20% water) and B (100% ethyl acetate). Gradient: 0 min; 20% B, 2.5 min; 22.5% B, 20–22.5 min; 50% B, 24–26 min; 80% B,

31–34 min; 100% B, 42–47 min; and 20% B, flow rate 1 mL min$^{-1}$. The calibration method was used for carotene quantification (mg g$^{-1}$ in DW).

### 2.5. Antioxidant Activity

Extracts were prepared by mixing about 0.05 g of dry lyophilized and homogenized plant material with 5 mL of 80% aqueous methanol. After 24 h in 4 °C temperature incubation, samples were centrifuged at 4000× $g$ for 15 min; the supernatant was collected for analysis. Antioxidant properties of plant material were evaluated as free radical scavenging activities and ferric-reducing antioxidant power (FRAP).

The ABTS (2,2′-azino-bis (3-ethylbenzothiazoline-6-sulphonic acid) radical scavenging activity. Radical cation was obtained by incubating the 7 mM ABTS stock solution (100 mL) with 2.45 mM potassium persulfate ($K_2S_2O_8$ 99% purity; Sigma-Aldrich, Burlington, MA, USA) and allowing the mixture to stand in the dark at room temperature for 12–16 h before use. After that, 20 μL of the prepared sample was mixed with 280 μL of ABTS solution (ABTS stock solution was diluted 1:7), and the absorbance was measured after 11 min (plateau phase) at 734 nm (SPECTROstar Nano, BMG Labtech microplate reader, Ortenberg, Germany). The ABTS scavenging activity of plant leaf extracts was calculated as the difference between the initial absorbance and after reacting for 10 min. A calibration curve was determined using Trolox (6-hydroxy-2,5,7,8-tetramethychroman-2-carboxylic acid; 97% purity; Sigma-Aldrich, Burlington, MA, USA) as an external standard with a range of concentrations from 0.1 to 0.8 mM ($R^2$ = 0.99). Results expressed as Trolox equivalent of ABTS scavenged per 1 g of dry weight (mmol TE g$^{-1}$ DW).

For DPPH (2-diphenyl-1-picrylhydrazyl) assay, a stable 126.8 μM DPPH (100% purity; Sigma-Aldrich, Burlington, MA, USA) solution was prepared in methanol. Subsequently, 280 μL of the DPPH solution was transferred to a test tube and mixed with 20 μL of the extract. The absorbance was scanned at 515 nm (SPECTROstar Nano, BMG Labtech microplate reader, Ortenberg, Germany) while reacting for 16 min. The free radical scavenging capacity was expressed as the Trolox equivalent of DPPH radicals scavenged per 1 g of dry plant weight (mmol TE g$^{-1}$ DW). A calibration curve was determined using Trolox (6-hydroxy-2,5,7,8-tetramethychroman-2-carboxylic acid; 97% purity; Sigma-Aldrich, Burlington, MA, USA) as an external standard with a range of concentrations from 0.1 to 0.6 mM ($R^2$ = 0.99).

The FRAP (Ferric reducing antioxidant power) method is based on reducing ferric ions ($Fe^{3+}$) to ferrous ions ($Fe^{2+}$). The fresh working solution was prepared by mixing 300 mM, pH 3.6 acetate buffer, 10 mM TPTZ (2,4,6-tripyridyl-s-triazine) solution in 40 mM HCl, and 20 mM $FeCl_3 \times 6H_2O$ at 10:1:1 ($v/v/v$). 20 μL of the sample was mixed with 280 μL of working solution and incubated in the dark for 30 min. Readings of the colored product (ferrous tripyridyl-triazine complex) were then taken at 593 nm (SPECTROstar Nano, BMG Labtech microplate reader, Ortenberg, Germany). A calibration curve was determined using $Fe_2(SO_4)_3$ (Iron (III) sulfate; 97% purity; Sigma-Aldrich, Burlington, MA, USA) as an external standard with a range of concentrations from 0.005 to 0.5 mM ($R^2$ = 0.99). The antioxidant power is expressed as $Fe^{2+}$ reduction antioxidant capacity ($Fe^{2+}$ μmol g$^{-1}$ DW).

### 2.6. Total Phenolic Content

Extracts were prepared by mixing about 0.05 g of dry lyophilized and homogenized plant material with 5 mL of 80% aqueous methanol. After 24 h in 4 °C temperature incubation, samples were centrifuged at 4000× $g$ for 15 min; the supernatant was collected for analysisThe total content of phenolic compounds was determined as gallic acid equivalents. A 20 μL aliquot of the sample extract was mixed with 20 μL of 10% ($w/v$) Folin–Ciocalteu reagent, 160 μL of 1 M $Na_2CO_3$ solution. After incubation for 20 min in the dark, the absorbance was measured at 765 nm (SPECTROstar Nano, BMG Labtech microplate reader, Ortenberg, Germany). The total phenolic compounds quantity mg g$^{-1}$ was calculated from the calibration curve of the gallic acid (0.01–0.1 mg mL$^{-1}$, $R^2$ = 0.99).

*2.7. Target Phenolic Compounds*

Chicoric, rosmarinic acids, and epicatechin were identified and quantified in 80% methanolic extracts by high-performance liquid chromatography (HPLC) method with diode array detection (DAD) at 280 nm on Shimadzu 10 A system (Shimadzu, Japan) and NUCLEODUR EC 150/4 3μm column (Macherey-Nagel, Dueren, Germany). Injection volume—10 μL. Mobile phase gradient consisted of A (acetonitrile) and B (1% acetic acid in water): 95% B for 25 min, followed by a linear gradient to 70% B at 5 min, then to 5% B in 2 min and hold for 1 min, then elevated till 95% B in 7 min, and hold until the 40 min. Flow rate—1 mL min$^{-1}$. The calibration method, using standard solutions, was used for phenolic compound quantification (mg g$^{-1}$ in plant DW).

*2.8. α-Tocopherol Contents*

α-Tocopherol contents were evaluated using the high-performance liquid chromatography (HPLC) method. Extracts were prepared by mixing about 0.05 g of dry lyophilized and homogenized plant material with 5 mL of hexane. An HPLC 10A system, equipped with an RF-10A fluorescence detector (Shimadzu, Japan) and Pinnacle II silica column, with 5 μm particle size, 150 mm, 4.6 mm (Restek, Pennsylvania, USA), was used for analysis. The mobile phase was 0.5% isopropanol (Merck, Germany) in n-hexane (isocratic elution); a flow rate of 1 mL min$^{-1}$. Peaks were detected using an excitation wavelength of 295 nm and an emission wavelength of 330 nm. The identification of α tocopherol was carried out by comparing the retention time and spectra of the peaks with the standard materials. The results are presented as α tocopherol contents mg g$^{-1}$ in plant DW.

*2.9. Statistical Analysis*

Statistical analysis was performed using Microsoft Excel 2016 and Addinsoft XLSTAT 2022 statistical and data analysis software (Long Island, NY, USA). Two-way analysis of variance (ANOVA) followed by Tukey's honestly significant difference test ($p < 0.05$) for multiple comparisons was used to evaluate differences between means ($n = 9$ (3 experimental $\times$ 3 analytical replications) of measurements. Multivariate principal component analysis (PCA) and correlation analysis was performed.

## 3. Results

UV-A supplement to the basic light spectrum positively affected the appearance of leafy vegetables. However, it did not have a pronounced impact on consumer opinion ratings (Figure 2).

The 405 nm and 385 nm supplemental light had different effects on the weight and leaf area of different leafy vegetables (Figure 3). There was no significant impact of investigated UV-A wavelengths on mustard and lettuce 'Afficion' plant green weight and leaf area. Meanwhile, 385 nm supplemental light reduced the green weight of kale and lettuce 'Belino' by 34 and 23.7% and leaf area by 48 and 31.8%, respectively, compared to basic lighting (Figures 2 and 3). Both UV-A wavelengths, 385 nm, and 405 nm, significantly decreased kale's green weight and leaf area (by 35.4% and 39.5%, respectively) compared to basic lighting.

Untrained consumer tests were performed to explore the main sensory attributes of leafy vegetables relevant to the consumer: appearance, crispness, taste (sweetness, bitterness), and overall assessment (Figure 4). According to consumer opinion, supplemental UV-A 385 nm light had no significant effect on the main evaluated properties in lettuces of different varieties (Figure 4c–e). However, supplemental violet 405 nm light resulted in increased bitterness of all three lettuce varieties from 3.0–3.1 up to 3.8–4.0 points. The 385 nm supplemental lighting elevated the bitterness of the kale (Figure 4b). However, supplemental violet 405 nm reduced the aftertaste of kale perceived by consumers from 3.8 to 2.8 points. UV-A had the greatest effect on the flavor characteristic of mustard compared to other plants (Figure 4a). The 385 nm supplemental lighting increased sweetness and

overall assessment appreciation in mustard. Also, both 405 nm and 385 nm increased mustard crispiness and aftertaste.

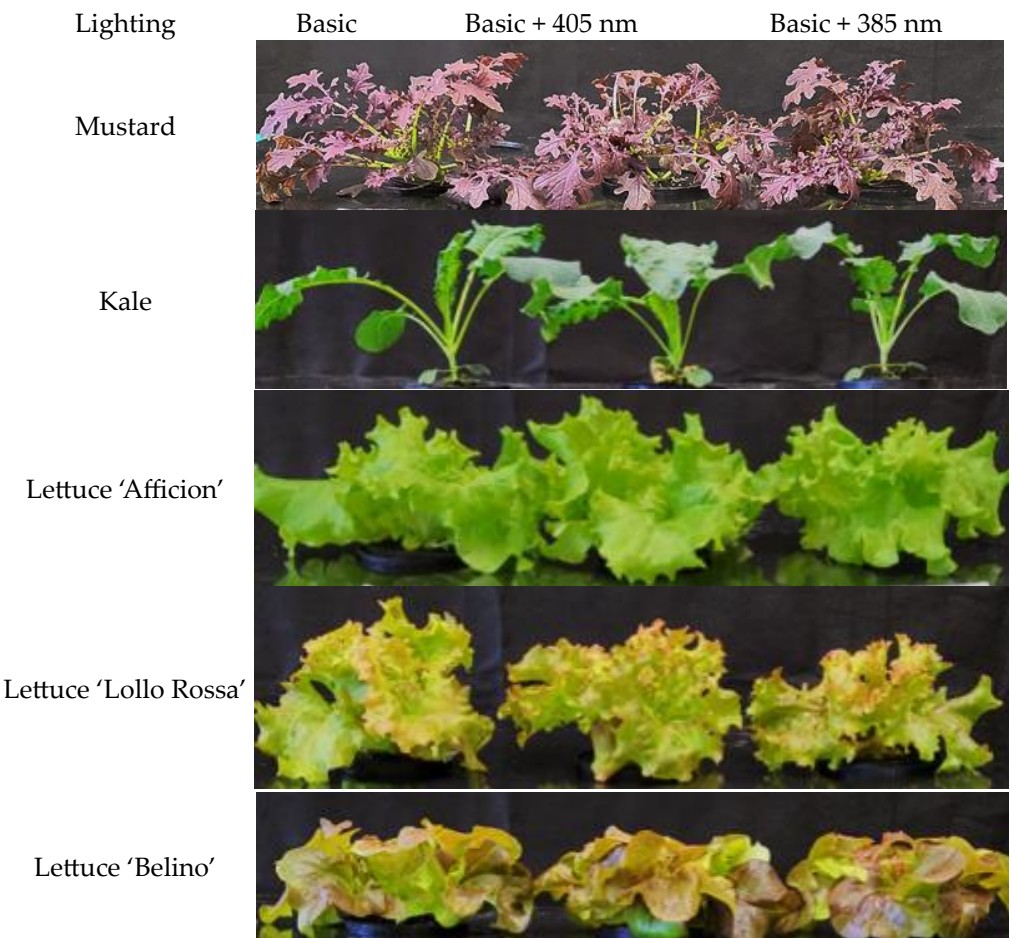

**Figure 2.** Effect of UV-A on the appearance of leafy vegetables.

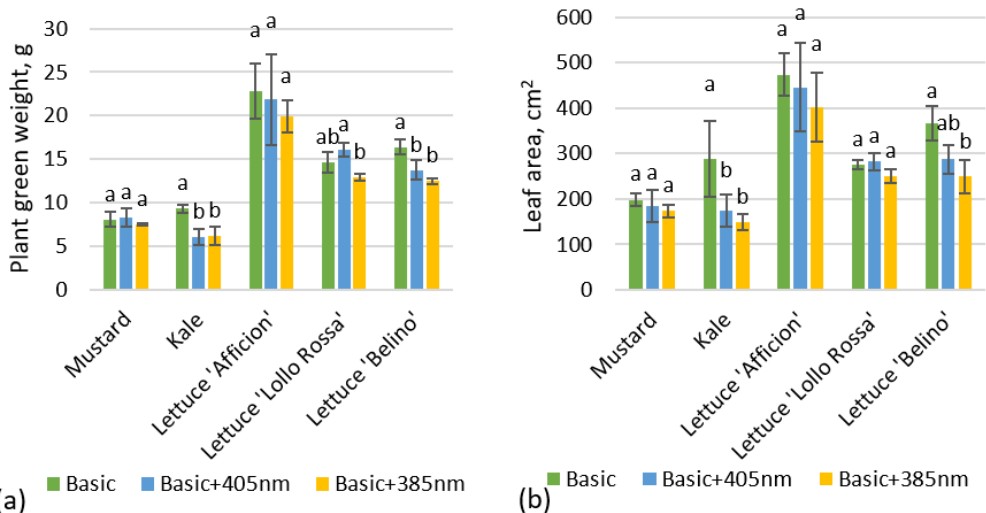

**Figure 3.** Effect of supplemental UV-A/violet light on leafy vegetables' green weight (**a**) and leaf area (**b**). Different letters above error bars indicate statistically significant differences between means of treatments within the plant species/varieties according to ANOVA Tukey's test when $p \leq 0.05$.

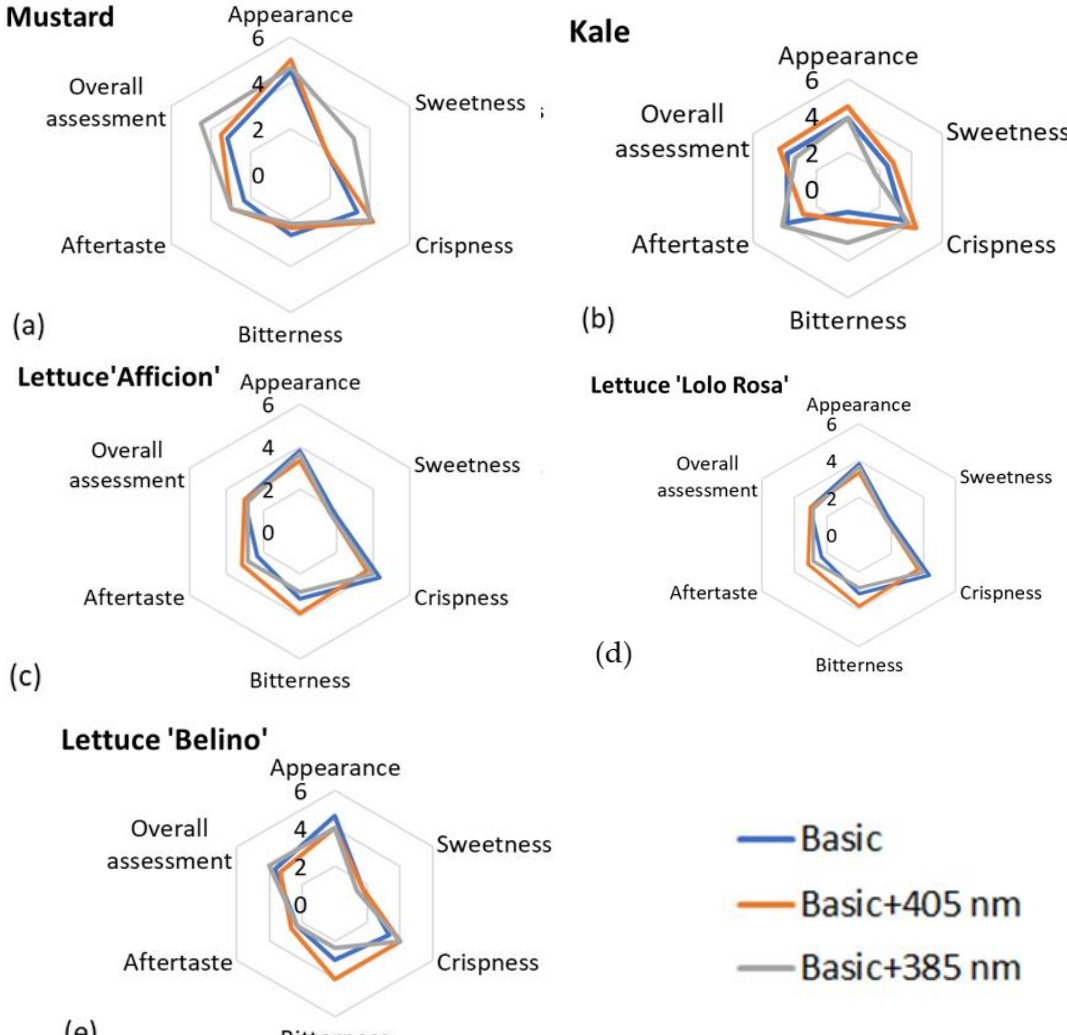

**Figure 4.** Sensory attributes of leafy vegetables mustard (**a**), kale (**b**), lettuces 'Afficion' (**c**), 'Lola Rosa' (**d**) and 'Belino' (**e**) cultivated under basic lighting, supplemented with UV-A and UV-A/violet light wavelengths.

No statistically significant correlation between sensory attributes and fructose, glucose, or other biochemical compound content was determined in investigated leafy vegetables (Table S1A–F). While consumers indicate pronounced differences in sweetness scores in mustard and kale, treated with different lighting (Figure 4), biochemical analysis results show different trends. Supplemental UV-A lighting significantly affected soluble sugar contents in kale, lettuce 'Afficion' and 'Belino' (Figure 5). Although no significant differences were determined in mustard and lettuce 'Lollo Rossa', treated with supplemental UV-A light, compared to basic lighting. Supplemental 385 nm lighting resulted in increased fructose (up to 3 times) but decreased glucose contents by 26% in kale, compared to basic lighting. 385 nm light also, had a negative impact on glucose accumulation in lettuce 'Belino'. Meanwhile, supplemental violet 405 nm light had a negative impact on fructose accumulation in lettuce 'Afficion' and 'Belino', up to 10% and 19% respectively.

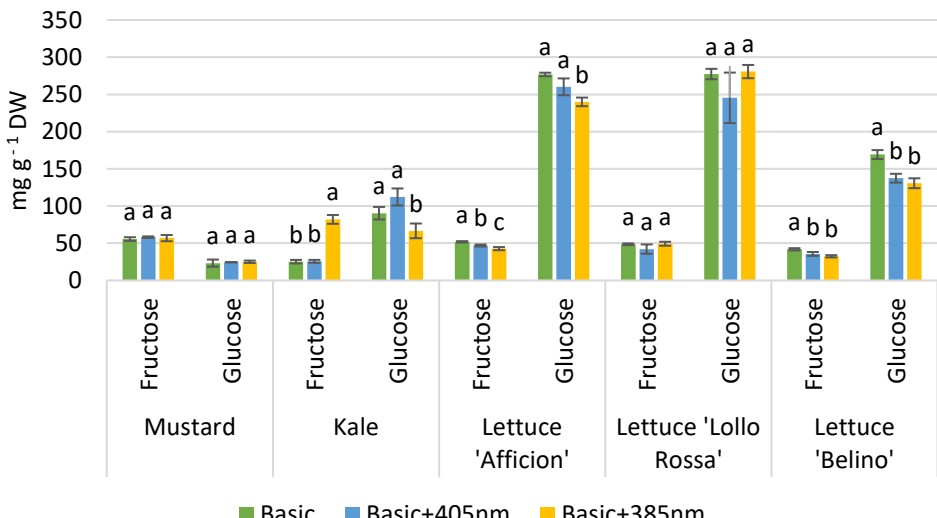

**Figure 5.** Effect of supplemental UV-A/violet light on the soluble sugar (fructose and glucose) contents in leafy vegetables. Different letters above error bars indicate statistically significant differences between means of treatments within the plant species/varieties according to ANOVA Tukey's test, when $p \leq 0.05$.

The response of the antioxidant system to UV-A and UV-A/violet light differed between plants. Both 405 nm and 385 nm supplemental light significantly increased DPPH and ABTS free radical scavenging activity in mustard (by 26%, 29%, and 34%, 43% compared to basic lighting, respectively) (Figure 6a). DPPH free radical scavenging activity decreased by 31% in kale under supplemental 385 nm lighting compared to basic lighting. Meanwhile, antioxidant activity according to DPPH free radical scavenging activity was determined to be significantly higher in lettuce 'Afficion' and 'Belino' under both supplemental UV-A and UV-A/violet lighting. In lettuce 'Lollo Rossa,' no significant impact of supplemental lighting was determined; however, FRAP antioxidant power was slightly increased under supplemental light exposure (Figure 6b). In mustard, DPPH and ABTS free radical scavenging activity and FRAP antioxidant power showed a statistically significant increase under both supplemental UV-A and violet LED wavelengths. 405 nm supplemental light resulted in 11–22% higher FRAP antioxidant power in all tested lettuce varieties than basic lighting. In comparison, 385 nm had no significant effect on this parameter in lettuce and kale (Figure 6b).

The effects of UV-A/violet supplemental LED lights on the content of phytochemical compounds in leafy vegetables differed by plant species (Figure 7). α + β carotene content was determined up to 2–3 times higher in lettuce 'Lollo Rossa' under supplemental violet and UV-A wavelengths, compared to basic spectra. Meanwhile, in lettuce 'Belino', α + β carotene content was determined to be 15–18% lower (Figure 7a). Supplemental violet and UV-A wavelengths also increased α-tocopherol content up to 1.6–3.8 times in all lettuce varieties, compared to basic lighting (Figure 7b). Supplemental lighting had no significant impact on mustard phytochemical contents, only UV-A 385 nm resulted in a higher amount of epicatechin (from 0.02 mg g$^{-1}$ in basic lighting treatment to 0.26 mg g$^{-1}$) (Figure 7c). Although, epicatechin also significantly increased in lettuces 'Afficion' and 'Belino' under supplemental 405 nm wavelength treatment (~30% higher, compared to basic lighting), but decreased by 23% under 385 nm wavelength in lettuce 'Belino' leaves (Figure 7c). 385 nm also reduced chicoric acid contents in lettuce and kale; it decreased by 10–32% compared to basic lighting (Figure 7d). Rosmarinic acid contents were determined lower only in kale under supplemental light. Meanwhile, significant differences were determined in rosmarinic acid contents in other leafy vegetables (Figure 7e).

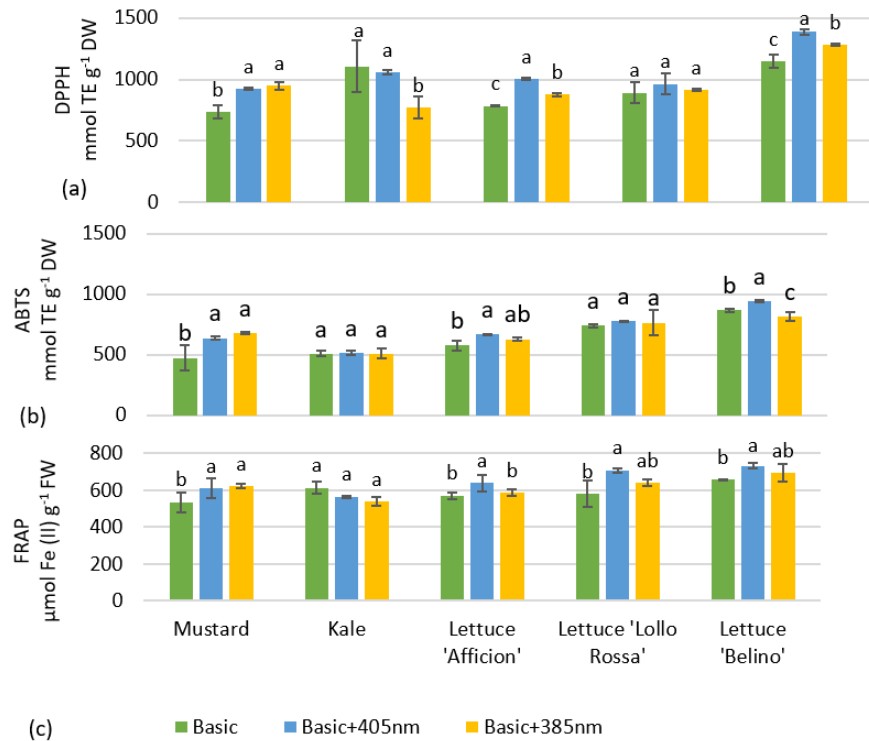

**Figure 6.** Effect of UV-A/violet supplemental light wavelengths on the DPPH (**a**), ABTS (**b**) free radical scavenging activity, and FRAP (**c**) antioxidant power in leafy vegetables. Different letters above error bars indicate statistically significant differences between means of treatments within the plant species/varieties according to ANOVA Tukey's test when $p \leq 0.05$.

A principal component analysis (PCA) show, that lighting impacts are diverse for different leafy vegetables and 385 nm and 405 nm supplemental light has a differential impact (Figure 8a–f). The best replication of the results between experimental repetitions was observed in kale and lettuce 'Belino' (Figure 8b,e). 385 and 405 nm light had significantly different impacts on these leafy vegetables too. The general PCA scatterplot of all plants (Figure 8e) indicates the primary differences between plant genotypes, partitioned into 3 groups: mustard and kale (both *Brassicaceae* vegetables), green leaf lettuce and red leaf lettuce. Mustard and kale were distinguished from other vegetables according to the F1 component. Factor loadings (Table S2) address these differences to the variation in fresh weight, leaf area, glucose contents, ABTS free radical scavenging activity, and chicoric acid contents. Lettuce varieties were divided into 2 groups (red leaf 'Belino' vs. green leaf 'Lollo Rossa' and 'Afficion') according to the F2 component. This, according factor loadings, can be addressed to DPPH free radical scavenging activity, FRAP, and epicatechin contents variation. 'Belino' lettuce is also distinguished by significant impact of UV-A/violet light on biomass parameters. Analyzing all plants separately, variation according to the F1 component in most of them can be addressed to the antioxidant properties (Figure 8a–e). Following that, 'Lollo Rossa' lettuce and kale tend to have higher antioxidant properties when illuminated with 405 nm supplemental light; mustard and 'Afficion' lettuce did not show a significant difference in antioxidant parameters between 385 and 405 nm supplemental light treatments, while 'Belino' lettuce performed the best under basic lighting. However, this can be assigned to the negative UV-A/violet light impact on biomass parameters. Distribution according to F2 component foremost can be assigned for sensory parameter variation (Table S2), and mustard, kale and 'Belino, 'Afficion' lettuces have leading scores when cultivated under 405 nm supplemental lighting.

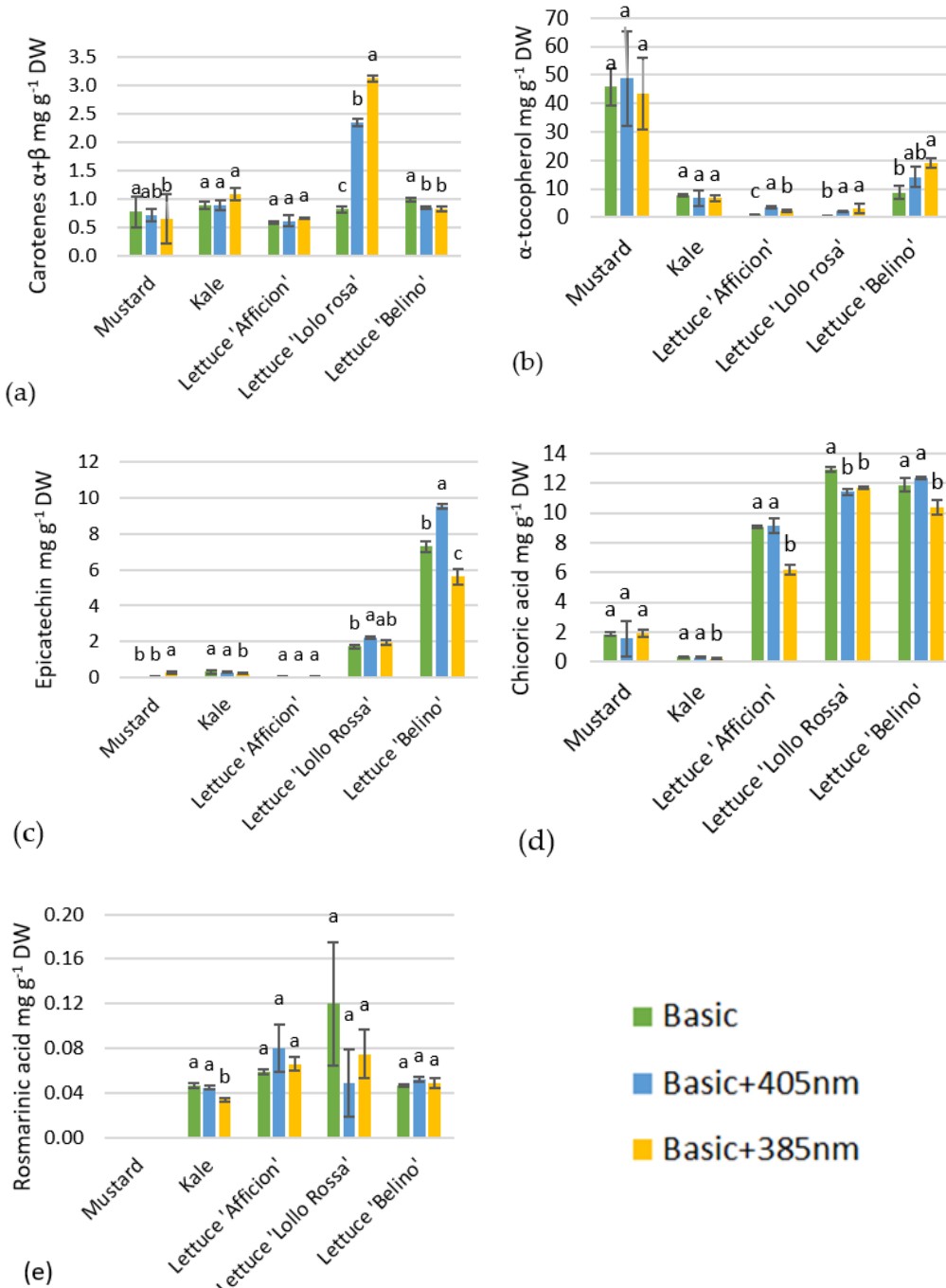

**Figure 7.** Effect of UV-A/violet supplemental light wavelengths on the amount of phytochemical compounds with nutritional value: α + β carotenes (**a**), α -tocopherol (**b**), Epicatechin (**c**), Chicoric acid (**d**) and Rosmarinic acid (**e**) in leafy vegetables. Different letters above error bars indicate statistically significant differences between means of treatments within the plant species/varieties according to ANOVA Tukey's test when $p \leq 0.05$.

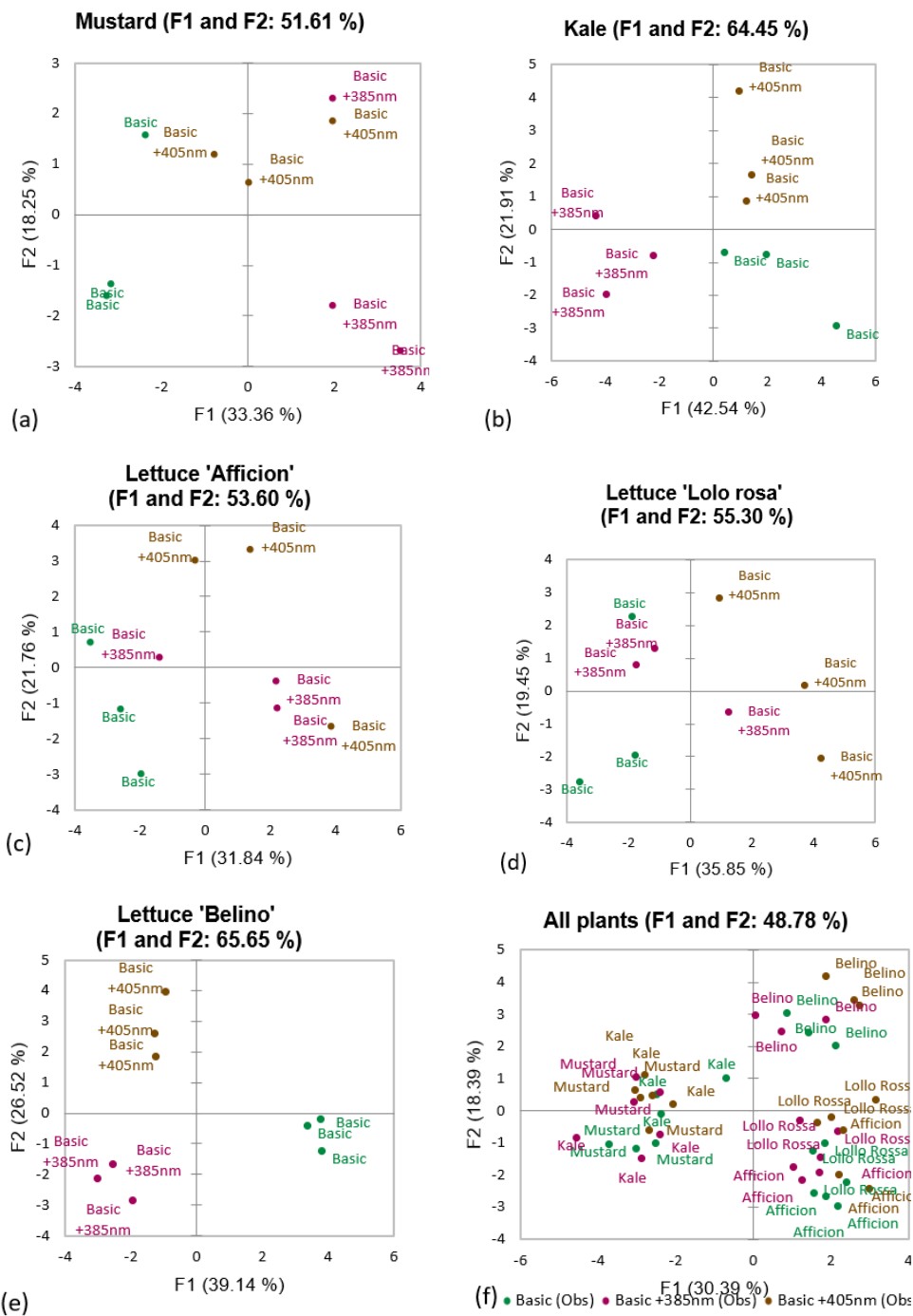

**Figure 8.** Principal Component Analysis (PCA) scatterplots indicating differences in lighting impact on individual plants (mustard (**a**); kale (**b**), lettuce 'Afficion' (**c**), lettuce 'Lollo Rossa' (**d**), lettuce 'Belino' (**e**), all plants (**f**) in one PCA. Different colors show different lighting treatments in all figures.

## 4. Discussion

Freshness, appearance, nutritional value, and flavor [27–29] are the primary sensory parameters affecting consumers' preferences. According to our research, leafy vegetables grown under supplemental UV-A/violet lighting did not have significant morphological and biomass differences compared to basic lighting (Figure 2). However, kale and 'Belino' lettuce were more sensitive to both 405 nm and 385 nm exposure, that resulted in reduced plant green weight and leaf area. According to biometric parameters, other investigated plants were tolerant to UV-A and UV-A/violet light exposure at applied irradiance of 1.1 mW cm$^{-1}$ (11 W m$^{-2}$). Other studies reported controversial results, suggesting

species, UV-A dosage, wavelength, and basic lighting-specific impacts. He et al. [30] compared the effects of 5, 10 and 15 W m$^{-2}$ 380 nm UV-A irradiance (basic light intensity of 250 µmol·m$^{-2}$·s$^{-1}$ for 12 h photoperiod) on kale root and above-ground part growth. They concluded that the kale growth rate falls when UV-A irradiance is higher than 10 W m$^{-2}$. However, Hu et al. [18] showed, that 385 nm light, applied only for a short 5-day period before kale harvest, 4 h per day, did not have growth limiting impact (basic light intensity of 250 µmol·m$^{-2}$·s$^{-1}$ for 12 h photoperiod) at an irradiance of 10 and 15 W m$^{-2}$. Choi et al. [17], demonstrated, that impact of UV-A, applied for 7 days pre-harvest at 30 W m$^{-2}$ (at 150 µmol m$^{-2}$ s$^{-1}$ basic lighting) had UV-A wavelength-specific impacts on kale growth. Exploring the effects of 365, 375, 385, 395 and 405 nm, it was concluded that treatments with 395 and 405 nm wavelengths increased most of the assessed growth characteristics and photosynthetic rates compared to the control and shorter UV-A wavelengths.

Comparable results were obtained with lettuce [15,31]: 10, 20 and 30 µmol m$^{-2}$ s$^{-1}$ of 365 nm UV-A (237 µmol m$^{-2}$ s$^{-1}$ basic lighting within 16 h photoperiod) resulted in increased shoot dry weight and leaf area, while moderate (20 µmol m$^{-2}$ s$^{-1}$) intensity had the most pronounced impact on growth parameters [15]. UV-A radiation substantially enhanced the accumulation of secondary metabolites, e.g., anthocyanin and ascorbic acid. In our study, supplemental UV-A/violet light was applied throughout the whole cultivation experiment. Therefore, prolonged UV-A impact, even at relatively low irradiance, resulted in some plants' slight leaf area and weight reduction. Supplemental 385 and 405 nm UV-A/violet LED light also evoked antioxidant response. DPPH free radical scavenging activity and FRAP antioxidant power were determined higher under both investigated wavelengths in mustard and lettuce 'Belino'; while in lettuce 'Lollo Rossa' and 'Afficion', positive impact on antioxidant properties was determined only under 405 nm light exposure. The higher antioxidant activity was not directly linked to higher contents of analyzed bioactive compounds; moreover, the light impacts on phytochemical contents were highly dependent on plant species. Under UV-A/violet exposure, α+β carotene contents were accumulated remarkably higher (up to 3 times) in lettuce 'Lollo Rossa'. α-tocopherol quantity was determined to be higher in all lettuce varieties. UV-A-induced changes in epicatechin contents were determined only in mustard, while rosmarinic and chicoric acid—were in kale. However, these differences in measured phytochemical contents, including sugars, do not correlate with sensory properties, suggesting that consumer preferences should be evaluated separately from the phytochemical values of investigated plants. Aftertaste and bitterness had the most differing scores between investigated lighting treatments. Other studies analyzing light spectrum impact on lettuce sensory properties confirm that bitterness and aftertaste are mainly associated with the changes in bioactive phytochemicals [32,33], but consumer preferences do not meet the phytochemical value of treated leafy vegetables and should be evaluated separately. In our study, 405 nm supplemental light resulted in a remarkably higher bitterness score in all investigated lettuce varieties. It coincides with higher antioxidant activities and higher epicatechin contents in their leaves. However, these results didn't show any significant correlation (Table S1). 'Belino' red leaf lettuce was more sensitive to UV-A exposure than investigated green leaf varieties. Lee et al. [22] treated green and red leaf lettuce cultivars with supplemental UV-A 375 nm light (~40 W m$^{-2}$) for 3–6 days and observed shoot dry mass decrease only at the longest UV-A treatment of 6 days. Under these conditions, it was concluded that green-leaf lettuce was more responsive to preharvest supplemental UV-A treatment than red-leaf lettuce: it accumulated~20% more proteins and higher amounts of Mg, Cu, and Zn compared to red-leaf lettuce. Nguyen et al. [34] performed experiments with red and green perilla cultivars and did not find significantly different responses of the cultivars to 365, 385, 415 nm light, 25 W m$^{-2}$ applied for 7 days. At our UV-A dosages applied, red leaf lettuce 'Belino' contained the highest amounts of α-tocopherol, epicatechin, however more pronounced reducing UV-A light impact was observed on its growth parameters and sugar (fructose and glucose contents, while growth parameters and sugar contents were not remarkably affected in other investigated leafy vegetables. Li et al. [35] state

that 380 nm and 400 nm UV-A wavelengths (at 40 W m$^{-2}$ irradiance) wavelengths did not have an effect on soluble sugar content in kale; however, Jiang et al. [19] showed that 380 nm UV-A light has a positive impact on soluble sugar contents in kale only at moderate, 12 μmol· m$^{-2}$s$^{-1}$ supplemental light exposure.

In summary, obtained results suggest that the value of supplemental UV-A light for the growth and quality of leafy vegetables cannot have the general assessment. Considering the UV-A irradiance and duration, the impacts on marketable parameters—plant growth, bioactive metabolite contents, and sensory scores should pass plant species and cultivar-specific evaluation. Under investigated, moderate UV-A/violet light dosage conditions, when supplemental UV-A is applied for the 12 h photoperiod and whole cultivation cycle, 405 nm light is preferable for higher antioxidant and sensory properties of kale, mustard, green leaf lettuce. However, consumer preferences do not meet the phytochemical value of investigated vegetables, therefore, lighting conditions for optimal nutrient contents in leafy vegetables and sensory properties should be assessed separately.

**Supplementary Materials:** The following supporting information can be downloaded at: https://www.mdpi.com/article/10.3390/horticulturae9050551/s1. Table S1. Pearson correlation matrix. Correlation coefficients represent relations between sensory properties and phytochemical contents in different leafy vegetables. Values in bold represent statistically significant correlations ($p \leq 0.05$). Table S2 Principal component analysis factor loadings for leafy vegetables.

**Author Contributions:** Conceptualization, A.V. and G.S.; methodology, A.V.; formal analysis, K.L. and R.S.; investigation, K.L. and R.S.; data curation, A.V., G.S. and A.B.; writing—original draft preparation, K.L.; writing—review and editing, A.V.; visualization, K.L and A.V.; supervision, A.V. All authors have read and agreed to the published version of the manuscript.

**Funding:** This research was funded by the European Regional Development Fund according to the supported activity 'Research Projects Implemented by World-class Researcher Groups' under Measure No. 01.2.2-LMT-K-718-01-0049.

**Institutional Review Board Statement:** Not applicable.

**Informed Consent Statement:** Not applicable.

**Data Availability Statement:** Not applicable.

**Conflicts of Interest:** The authors declare no conflict of interest.

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
