# Peer review of "UV-A for Tailoring the Nutritional Value and Sensory Properties of Leafy Vegetables"

_horticulturae, doi:10.3390/horticulturae9050551_

Round 1

Reviewer 1 Report

The manuscript entitled “UV-A for improving the nutritional quality of leafy vegetables” by K. LaužikÄ— et al. has been reviewed.

Comment 1

The title of the MS needs to be modified to reflect the content of the study.

Comment 2

The MM section should be rewritten and improved with detailed information. I would suggest that authors add a number before subtitle in the each section (2.1 Plant material and growing conditions; 2.2 ...). For example in Section 2.1., I would suggest that authors start with a description of the samples and then described the experimental lighting design. What is the present picture below Table 1?

Comment 3

I would suggest that the authors don’t start sentences with a number or abbreviation. For example lines 17, 99, 207, 226 and others.

Comment 4

The authors should provide more data about the consumer panelists included in the study. Why did the authors choose untrained consumers?

Comment 5

All references in the MS should be uniforms.

Comment 6

Line 136: It should be divided into two subtitles, antioxidant activities and total phenolic content. How authors determined total phenolic content?

Comment 7

I would suggest that authors also add a) and b) in Figure 2 (Above left corner- all figures should be the same).  Please check that the size of the words, and letters are the same in the figure. In line 255: Fig 5A in the Figure is present like (a). It should be changed.

Comment 8

Figure 4: mgg-1 change to mg g-1

p – should be change to italic p.

Comment 9

Line 266: Please delete one dot.

Comment 10

Could you please check where are the results expressed in the DW and where FW ? For example, in Line 122: sugar quantification (mg g-1 in dry plant weight (DW) and in Figure 4 is mgg-1 FW? FW – freight weight (FW).

Comment 11

Could you please improve and rewrite the section about a principal component analysis (PCA), it can be confusing for readers. I would suggest adding the principal component analysis biplot diagrams. What show 7f ? Please avoid overlapping text or dismissed letters. I would suggest that authors add toolbars to decrease text in the Figure.

Comment 12

Line 327: „ Consumer have high expecations for the quality of leady vegetables“-  Please delete or define. The sentence lack precision.

Line 376-377- rewrite this sentence.

Author Response

Thank you for Your review

Comment 1

The title of the MS needs to be modified to reflect the content of the study.

We changed the title:

UV-A for tailoring the nutritional value and sensory attributes of leafy vegetables

Comment 2

The MM section should be rewritten and improved with detailed information. I would suggest that authors add a number before subtitle in the each section (2.1 Plant material and growing conditions; 2.2 ...). For example in Section 2.1., I would suggest that authors start with a description of the samples and then described the experimental lighting design. What is the present picture below Table 1?

The structure of MM section is in agreement with journal guidelines. Authors choose leaving the lighting design to be introduced first, because it follows workflow: first lighting exposure, then sample collection.

Comment 3

I would suggest that the authors don’t start sentences with a number or abbreviation. For example lines 17, 99, 207, 226 and others.

Corrected

Comment 4

The authors should provide more data about the consumer panelists included in the study. Why did the authors choose untrained consumers?

The untrained consumer is the final recipient of the product (leafy vegetables), so the sensory properties of the plants are oriented towards what the average customer would buy.

Comment 5

All references in the MS should be uniforms.

Thank you for the comment, we tryed to uniform all references in the MS.

Comment 6

Line 136: It should be divided into two subtitles, antioxidant activities and total phenolic content. How authors determined total phenolic content?

Thank You for the comment, we supplemented MM section with Total phenol content determination method.

Comment 7

I would suggest that authors also add a) and b) in Figure 2 (Above left corner- all figures should be the same).  Please check that the size of the words, and letters are the same in the figure. In line 255: Fig 5A in the Figure is present like (a). It should be changed.

Thank You. All figures were unified and corrections in the text made.

Comment 8

Figure 4: mgg-1 change to mg g-1

p – should be change to italic p.

Thank You, corrections were made.

Comment 9

Line 266: Please delete one dot.

Deleted.

Comment 10

Could you please check where are the results expressed in the DW and where FW ? For example, in Line 122: sugar quantification (mg g-1 in dry plant weight (DW) and in Figure 4 is mgg-1 FW? FW – freight weight (FW).

Thank You for the comment, yes, there were some mistakes, we corrected it.

Comment 11

Could you please improve and rewrite the section about a principal component analysis (PCA), it can be confusing for readers. I would suggest adding the principal component analysis biplot diagrams. What show 7f ? Please avoid overlapping text or dismissed letters. I would suggest that authors add toolbars to decrease text in the Figure.

Section was rewriten, figures corrected.

Comment 12

Line 327: „ Consumer have high expecations for the quality of leady vegetables“-  Please delete or define. The sentence lack precision.

We desided to delete it, next sentence says almost the same.

Line 376-377- rewrite this sentence.

Rewrited.

Reviewer 2 Report

Point1: Lines 109-110: What is the average age and gender ratio of the consumer panellists?

Point2: Line 199: Does "n=9 (3 experimental x 3 analytical replications)" refer to biological replicates and technical replicates separately?

Point3: Line 209: What specific plant organs or parts does "plant green weight" refer to? It is suggested to specify the organ or use a more specific term.

Point4: Line 222: How were sweetness, crispness, and aftertaste evaluated? Were they directly consumed?

Point5: Line 225: Which sensory property does bitterness belong to?

Point6: What software was used to process the data in Figure 3?

Point7: What software was used for PCA analysis?

Point8: The light and temperature for the three plants growth and development set under the same conditions by the authors. Do the three species have similar requirements for light, temperature, and other factors?

Point9: The author needs to modify the handling of the chart to match its characteristics.

Author Response

Thank you for Your review.

Point1: Lines 109-110: What is the average age and gender ratio of the consumer panellists?

Female and male ratio was 1:1, explained in line 111.

 Point2: Line 199: Does "n=9 (3 experimental x 3 analytical replications)" refer to biological replicates and technical replicates separately?

Three analytical replicates were made from each biological replicate and statistically evaluated together. 

Point3: Line 209: What specific plant organs or parts does "plant green weight" refer to? It is suggested to specify the organ or use a more specific term.

The green weight of the plant covers the entire plant, harvested at the root neck. This is the part that is served to the consumer.

Point4: Line 222: How were sweetness, crispness, and aftertaste evaluated? Were they directly consumed?

Consumers were provided with freshly harvested plants without specifying their growing conditions. Each user viewed the plants and tasted them. We served one type of plant at a time.

Point5: Line 225: Which sensory property does bitterness belong to?

Bitterness, together with sweetness, represents taste, as the sensory attribute. Explained in line 222-223.

Point6: What software was used to process the data in Figure 3?

Microsoft Excel was used to process the data in figure 3.

Point7: What software was used for PCA analysis?

Addinsoft XLSTAT 2022 statistical and data analysis software as indicated in statistical analysis method section.
Point8: The light and temperature for the three plants growth and development set under the same conditions by the authors. Do the three species have similar requirements for light, temperature, and other factors?

Point9: The author needs to modify the handling of the chart to match its characteristics.

Thank You for the coment. We check all handling to match it to charts.

Round 2

Reviewer 1 Report

The authors have improved the MS well. I recommend accepting after the minor correction the MS UV-A for tailoring the nutritional value and sensory properties of leaf vegetables“.

·        it should be same °C

·        +4 °C change to 4°C

·        line 188: “analysis. The total”

·        shoud be in the subscript Na2CO3

·        Line 195: should be ml min-1

·        Figure 7 ...please add (e)

·        The letters in every figure should be in the left corner above the figure not below. I would suggest that authors changed that.

Author Response

We greatly appreciate your comments, which have helped to improve the quality of our manuscript.

Corrections have been made.